# Actor-Critic Algorithms for Risk-Sensitive MDPs

**Prashanth L.A.**
INRIA Lille - Team SequeL

**Mohammad Ghavamzadeh**[*]
INRIA Lille - Team SequeL & Adobe Research

## Abstract

In many sequential decision-making problems we may want to manage risk by minimizing some measure of variability in rewards in addition to maximizing a standard criterion. Variance-related risk measures are among the most common risk-sensitive criteria in finance and operations research. However, optimizing many such criteria is known to be a hard problem. In this paper, we consider both discounted and average reward Markov decision processes. For each formulation, we first define a measure of variability for a policy, which in turn gives us a set of risk-sensitive criteria to optimize. For each of these criteria, we derive a formula for computing its gradient. We then devise actor-critic algorithms for estimating the gradient and updating the policy parameters in the ascent direction. We establish the convergence of our algorithms to locally risk-sensitive optimal policies. Finally, we demonstrate the usefulness of our algorithms in a traffic signal control application.

## 1 Introduction

The usual optimization criteria for an infinite horizon Markov decision process (MDP) are the *expected sum of discounted rewards* and the *average reward*. Many algorithms have been developed to maximize these criteria both when the model of the system is known (planning) and unknown (learning). These algorithms can be categorized to value function based methods that are mainly based on the two celebrated dynamic programming algorithms *value iteration* and *policy iteration*; and policy gradient methods that are based on updating the policy parameters in the direction of the gradient of a performance measure (the value function of the initial state or the average reward). However in many applications, we may prefer to minimize some measure of *risk* as well as maximizing a usual optimization criterion. In such cases, we would like to use a criterion that incorporates a penalty for the *variability* induced by a given policy. This variability can be due to two types of uncertainties: **1)** uncertainties in the model parameters, which is the topic of *robust* MDPs (e.g., [12, 7, 24]), and **2)** the inherent uncertainty related to the stochastic nature of the system, which is the topic of *risk-sensitive* MDPs (e.g., [10]).

In risk-sensitive sequential decision-making, the objective is to maximize a risk-sensitive criterion such as the expected exponential utility [10], a variance-related measure [19, 8], or the percentile performance [9]. The issue of how to construct such criteria in a manner that will be both conceptually meaningful and mathematically tractable is still an open question. Although risk-sensitive sequential decision-making has a long history in operations research and finance, it has only recently grabbed attention in the machine learning community. This is why most of the work on this topic (including those mentioned above) has been in the context of MDPs (when the model is known) and much less work has been done within the reinforcement learning (RL) framework. In risk-sensitive RL, we can mention the work by Borkar [4, 5] who considered the expected exponential utility and the one by Tamar et al. [22] on several variance-related measures. Tamar et al. [22] study stochastic shortest path problems, and in this context, propose a policy gradient algorithm for maximizing several risk-sensitive criteria that involve both the expectation and variance of the *return* random variable (defined as the sum of rewards received in an episode).

---

[*]Mohammad Ghavamzadeh is at Adobe Research, on leave of absence from INRIA Lille - Team SequeL.

In this paper, we develop actor-critic algorithms for optimizing variance-related risk measures in both discounted and average reward MDPs. Our contributions can be summarized as follows:

• In the discounted reward setting we define the measure of variability as the *variance of the return* (similar to [22]). We formulate a constrained optimization problem with the aim of maximizing the mean of the return subject to its variance being bounded from above. We employ the Lagrangian relaxation procedure [1] and derive a formula for the gradient of the Lagrangian. Since this requires the gradient of the value function at every state of the MDP (see the discussion in Sections 3 and 4), we estimate the gradient of the Lagrangian using two simultaneous perturbation methods: *simultaneous perturbation stochastic approximation* (SPSA) [20] and *smoothed functional* (SF) [11], resulting in two separate discounted reward actor-critic algorithms.[1]

• In the average reward formulation, we first define the measure of variability as the *long-run variance* of a policy, and using a constrained optimization problem similar to the discounted case, derive an expression for the gradient of the Lagrangian. We then develop an actor-critic algorithm with compatible features [21, 13] to estimate the gradient and to optimize the policy parameters.

• Using the ordinary differential equations (ODE) approach, we establish the asymptotic convergence of our algorithms to locally risk-sensitive optimal policies. Further, we demonstrate the usefulness of our algorithms in a traffic signal control problem.

In comparison to [22], which is the closest related work, we would like to remark that while the authors there develop policy gradient methods for stochastic shortest path problems, we devise actor-critic algorithms for both discounted and average reward settings. Moreover, we note the difficulty in the discounted formulation that requires to estimate the gradient of the value function at every state of the MDP, and thus, motivated us to employ simultaneous perturbation techniques.

## 2   Preliminaries

We consider problems in which the agent's interaction with the environment is modeled as a MDP. A MDP is a tuple $(\mathcal{X}, \mathcal{A}, R, P, P_0)$ where $\mathcal{X} = \{1, \ldots, n\}$ and $\mathcal{A} = \{1, \ldots, m\}$ are the state and action spaces; $R(x, a)$ is the reward random variable whose expectation is denoted by $r(x, a) = \mathbb{E}[R(x, a)]$; $P(\cdot|x, a)$ is the transition probability distribution; and $P_0(\cdot)$ is the initial state distribution. We also need to specify the rule according to which the agent selects actions at each state. A *stationary policy* $\mu(\cdot|x)$ is a probability distribution over actions, conditioned on the current state. In policy gradient and actor-critic methods, we define a class of parameterized stochastic policies $\{\mu(\cdot|x; \theta), x \in \mathcal{X}, \theta \in \Theta \subseteq \mathbb{R}^{\kappa_1}\}$, estimate the gradient of a performance measure w.r.t. the policy parameters $\theta$ from the observed system trajectories, and then improve the policy by adjusting its parameters in the direction of the gradient. Since in this setting a policy $\mu$ is represented by its $\kappa_1$-dimensional parameter vector $\theta$, policy dependent functions can be written as a function of $\theta$ in place of $\mu$. So, we use $\mu$ and $\theta$ interchangeably in the paper.

We denote by $d^\mu(x)$ and $\pi^\mu(x, a) = d^\mu(x)\mu(a|x)$ the stationary distribution of state $x$ and state-action pair $(x, a)$ under policy $\mu$, respectively. In the discounted formulation, we also define the discounted visiting distribution of state $x$ and state-action pair $(x, a)$ under policy $\mu$ as $d^\mu_\gamma(x|x^0) = (1 - \gamma) \sum_{t=0}^\infty \gamma^t \Pr(x_t = x|x_0 = x^0; \mu)$ and $\pi^\mu_\gamma(x, a|x^0) = d^\mu_\gamma(x|x^0)\mu(a|x)$.

## 3   Discounted Reward Setting

For a given policy $\mu$, we define the return of a state $x$ (state-action pair $(x, a)$) as the sum of discounted rewards encountered by the agent when it starts at state $x$ (state-action pair $(x, a)$) and then follows policy $\mu$, i.e.,

$$D^\mu(x) = \sum_{t=0}^\infty \gamma^t R(x_t, a_t) \mid x_0 = x, \ \mu, \qquad D^\mu(x, a) = \sum_{t=0}^\infty \gamma^t R(x_t, a_t) \mid x_0 = x, \ a_0 = a, \ \mu.$$

The expected value of these two random variables are the value and action-value functions of policy $\mu$, i.e., $V^\mu(x) = \mathbb{E}[D^\mu(x)]$ and $Q^\mu(x, a) = \mathbb{E}[D^\mu(x, a)]$. The goal in the standard discounted reward formulation is to find an optimal policy $\mu^* = \arg\max_\mu V^\mu(x^0)$, where $x^0$ is the initial state of the system. This can be easily extended to the case that the system has more than one initial state $\mu^* = \arg\max_\mu \sum_{x \in \mathcal{X}} P_0(x)V^\mu(x)$.

The most common measure of the *variability* in the stream of rewards is the *variance of the return*

$$\Lambda^{\mu}(x) = \mathbb{E}\big[D^{\mu}(x)^2\big] - V^{\mu}(x)^2 = U^{\mu}(x) - V^{\mu}(x)^2, \tag{1}$$

first introduced by Sobel [19]. Note that $U^{\mu}(x) \overset{\triangle}{=} \mathbb{E}\big[D^{\mu}(x)^2\big]$ is the *square reward value function* of state $x$ under policy $\mu$. Although $\Lambda^{\mu}$ of (1) satisfies a Bellman equation, unfortunately, it lacks the monotonicity property of dynamic programming (DP), and thus, it is not clear how the related risk measures can be optimized by standard DP algorithms [19]. This is why policy gradient and actor-critic algorithms are good candidates to deal with this risk measure. We consider the following risk-sensitive measure for discounted MDPs: for a given $\alpha > 0$,

$$\max_{\theta} V^{\theta}(x^0) \qquad \text{subject to} \qquad \Lambda^{\theta}(x^0) \leq \alpha. \tag{2}$$

To solve (2), we employ the Lagrangian relaxation procedure [1] to convert it to the following unconstrained problem:

$$\max_{\lambda} \min_{\theta} \Big( L(\theta, \lambda) \overset{\triangle}{=} -V^{\theta}(x^0) + \lambda\big(\Lambda^{\theta}(x^0) - \alpha\big)\Big), \tag{3}$$

where $\lambda$ is the Lagrange multiplier. The goal here is to find the saddle point of $L(\theta, \lambda)$, i.e., a point $(\theta^*, \lambda^*)$ that satisfies $L(\theta, \lambda^*) \geq L(\theta^*, \lambda^*) \geq L(\theta^*, \lambda), \forall \theta, \forall \lambda > 0$. This is achieved by descending in $\theta$ and ascending in $\lambda$ using the gradients $\nabla_{\theta} L(\theta, \lambda) = -\nabla_{\theta} V^{\theta}(x^0) + \lambda \nabla_{\theta} \Lambda^{\theta}(x^0)$ and $\nabla_{\lambda} L(\theta, \lambda) = \Lambda^{\theta}(x^0) - \alpha$, respectively. Since $\nabla \Lambda^{\theta}(x^0) = \nabla U^{\theta}(x^0) - 2V^{\theta}(x^0)\nabla V^{\theta}(x^0)$, in order to compute $\nabla \Lambda^{\theta}(x^0)$, we need to calculate $\nabla U^{\theta}(x^0)$ and $\nabla V^{\theta}(x^0)$. From the Bellman equation of $\Lambda^{\mu}(x)$, proposed by Sobel [19], it is straightforward to derive Bellman equations for $U^{\mu}(x)$ and the *square reward action-value function* $W^{\mu}(x, a) \overset{\triangle}{=} \mathbb{E}\big[D^{\mu}(x, a)^2\big]$ (see Appendix B.1 of [17]). Using these definitions and notations we are now ready to derive expressions for the gradient of $V^{\theta}(x^0)$ and $U^{\theta}(x^0)$ that are the main ingredients in calculating $\nabla_{\theta} L(\theta, \lambda)$.

**Lemma 1** *Assuming for all $(x, a)$, $\mu(a|x; \theta)$ is continuously differentiable in $\theta$, we have*

$$(1 - \gamma)\nabla V^{\theta}(x^0) = \sum_{x,a} \pi_{\gamma}^{\theta}(x, a|x^0)\nabla \log \mu(a|x; \theta)Q^{\theta}(x, a),$$

$$(1 - \gamma^2)\nabla U^{\theta}(x^0) = \sum_{x,a} \widetilde{\pi}_{\gamma}^{\theta}(x, a|x^0)\nabla \log \mu(a|x; \theta)W^{\theta}(x, a)$$

$$+ 2\gamma \sum_{x,a,x'} \widetilde{\pi}_{\gamma}^{\theta}(x, a|x^0)P(x'|x, a)r(x, a)\nabla V^{\theta}(x'),$$

*where $\widetilde{\pi}_{\gamma}^{\theta}(x, a|x^0) = \widetilde{d}_{\gamma}^{\theta}(x|x^0)\mu(a|x)$ and $\widetilde{d}_{\gamma}^{\theta}(x|x^0) = (1 - \gamma^2)\sum_{t=0}^{\infty} \gamma^{2t} \Pr(x_t = x|x_0 = x^0; \theta)$.*

The proof of the above lemma is available in Appendix B.2 of [17]. It is challenging to devise an efficient method to estimate $\nabla_{\theta} L(\theta, \lambda)$ using the gradient formulas of Lemma 1. This is mainly because **1)** two different sampling distributions ($\pi_{\gamma}^{\theta}$ and $\widetilde{\pi}_{\gamma}^{\theta}$) are used for $\nabla V^{\theta}(x^0)$ and $\nabla U^{\theta}(x^0)$, and **2)** $\nabla V^{\theta}(x')$ appears in the second sum of $\nabla U^{\theta}(x^0)$ equation, which implies that we need to estimate the gradient of the value function $V^{\theta}$ at every state of the MDP. These are the main motivations behind using simultaneous perturbation methods for estimating $\nabla_{\theta} L(\theta, \lambda)$ in Section 4.

## 4 Discounted Reward Algorithms

In this section, we present actor-critic algorithms for optimizing the risk-sensitive measure (2) that are based on two simultaneous perturbation methods: *simultaneous perturbation stochastic approximation* (SPSA) and *smoothed functional* (SF) [3]. The idea in these methods is to estimate the gradients $\nabla V^{\theta}(x^0)$ and $\nabla U^{\theta}(x^0)$ using two simulated trajectories of the system corresponding to policies with parameters $\theta$ and $\theta^+ = \theta + \beta\Delta$. Here $\beta > 0$ is a positive constant and $\Delta$ is a perturbation random variable, i.e., a $\kappa_1$-vector of independent Rademacher (for SPSA) and Gaussian $\mathcal{N}(0, 1)$ (for SF) random variables. In our actor-critic algorithms, the critic uses linear approximation for the value and square value functions, i.e., $\widehat{V}(x) \approx v^{\top}\phi_v(x)$ and $\widehat{U}(x) \approx u^{\top}\phi_u(x)$, where the features $\phi_v(\cdot)$ and $\phi_u(\cdot)$ are from low-dimensional spaces $\mathbb{R}^{\kappa_2}$ and $\mathbb{R}^{\kappa_3}$, respectively.

**SPSA**-based gradient estimates were first proposed in [20] and have been widely studied and found to be highly efficient in various settings, especially those involving high-dimensional parameters. The SPSA-based estimate for $\nabla V^{\theta}(x^0)$, and similarly for $\nabla U^{\theta}(x^0)$, is given by:

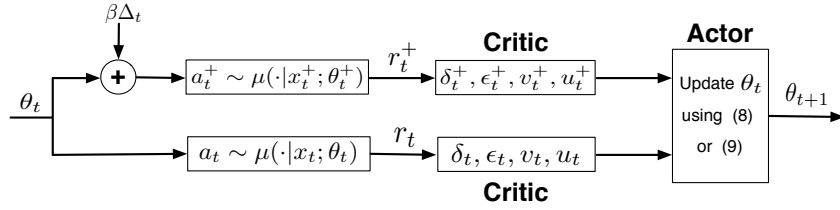

Figure 1: The overall flow of our simultaneous perturbation based actor-critic algorithms.

$$\partial_{\theta^{(i)}} \widehat{V}^\theta(x^0) \quad \approx \quad \frac{\widehat{V}^{\theta+\beta\Delta}(x^0) - \widehat{V}^\theta(x^0)}{\beta\Delta^{(i)}}, \qquad i = 1, \ldots, \kappa_1, \tag{4}$$

where $\Delta$ is a vector of independent Rademacher random variables. The advantage of this estimator is that it perturbs all directions at the same time (the numerator is identical in all $\kappa_1$ components). So, the number of function measurements needed for this estimator is always two, independent of the dimension $\kappa_1$. However, unlike the SPSA estimates in [20] that use two-sided balanced estimates (simulations with parameters $\theta - \beta\Delta$ and $\theta + \beta\Delta$), our gradient estimates are one-sided (simulations with parameters $\theta$ and $\theta + \beta\Delta$) and resemble those in [6]. The use of one-sided estimates is primarily because the updates of the Lagrangian parameter $\lambda$ require a simulation with the running parameter $\theta$. Using a balanced gradient estimate would therefore come at the cost of an additional simulation (the resulting procedure would then require three simulations), which we avoid by using one-sided gradient estimates.

**SF**-based method estimates not the gradient of a function $H(\theta)$ itself, but rather the convolution of $\nabla H(\theta)$ with the Gaussian density function $\mathcal{N}(\mathbf{0}, \beta^2 \boldsymbol{I})$, i.e.,

$$C_\beta H(\theta) = \int \mathcal{G}_\beta(\theta - z)\nabla_z H(z)dz = \int \nabla_z \mathcal{G}_\beta(z)H(\theta - z)dz = \frac{1}{\beta}\int -z'\mathcal{G}_1(z')H(\theta - \beta z')dz',$$

where $\mathcal{G}_\beta$ is a $\kappa_1$-dimensional probability density function. The first equality above follows by using integration by parts and the second one by using the fact that $\nabla_z \mathcal{G}_\beta(z) = \frac{-z}{\beta^2}\mathcal{G}_\beta(z)$ and by substituting $z' = z/\beta$. As $\beta \to 0$, it can be seen that $C_\beta H(\theta)$ converges to $\nabla_\theta H(\theta)$ (see Chapter 6 of [3]). Thus, a one-sided SF estimate of $\nabla V^\theta(x^0)$ is given by

$$\partial_{\theta^{(i)}} \widehat{V}^\theta(x^0) \quad \approx \quad \frac{\Delta^{(i)}}{\beta}\left(\widehat{V}^{\theta+\beta\Delta}(x^0) - \widehat{V}^\theta(x^0)\right), \qquad i = 1, \ldots, \kappa_1, \tag{5}$$

where $\Delta$ is a vector of independent Gaussian $\mathcal{N}(0, 1)$ random variables.

The overall flow of our proposed actor-critic algorithms is illustrated in Figure 1 and involves the following main steps at each time step $t$:
**(1)** Take action $a_t \sim \mu(\cdot|x_t; \theta_t)$, observe the reward $r(x_t, a_t)$ and next state $x_{t+1}$ in the first trajectory.
**(2)** Take action $a_t^+ \sim \mu(\cdot|x_t^+; \theta_t^+)$, observe the reward $r(x_t^+, a_t^+)$ and next state $x_{t+1}^+$ in the second trajectory.
**(3) Critic Update:** Calculate the temporal difference (TD)-errors $\delta_t, \delta_t^+$ for the value and $\epsilon_t, \epsilon_t^+$ for the square value functions using (7), and update the critic parameters $v_t, v_t^+$ for the value and $u_t, u_t^+$ for the square value functions as follows:

$$\begin{aligned} v_{t+1} &= v_t + \zeta_3(t)\delta_t\phi_v(x_t), & v_{t+1}^+ &= v_t^+ + \zeta_3(t)\delta_t^+\phi_v(x_t^+), \\ u_{t+1} &= u_t + \zeta_3(t)\epsilon_t\phi_u(x_t), & u_{t+1}^+ &= u_t^+ + \zeta_3(t)\epsilon_t^+\phi_u(x_t^+), \end{aligned} \tag{6}$$

where the TD-errors $\delta_t, \delta_t^+, \epsilon_t, \epsilon_t^+$ in (6) are computed as

$$\begin{aligned} \delta_t &= r(x_t, a_t) + \gamma v_t^\top \phi_v(x_{t+1}) - v_t^\top \phi_v(x_t), & \delta_t^+ &= r(x_t^+, a_t^+) + \gamma v_t^{+\top} \phi_v(x_{t+1}^+) - v_t^{+\top} \phi_v(x_t^+), \\ \epsilon_t &= r(x_t, a_t)^2 + 2\gamma r(x_t, a_t)v_t^\top \phi_v(x_{t+1}) + \gamma^2 u_t^\top \phi_u(x_{t+1}) - u_t^\top \phi_u(x_t), \\ \epsilon_t^+ &= r(x_t^+, a_t^+)^2 + 2\gamma r(x_t^+, a_t^+)v_t^{+\top} \phi_v(x_{t+1}^+) + \gamma^2 u_t^{+\top} \phi_u(x_{t+1}^+) - u_t^{+\top} \phi_u(x_t^+). \end{aligned} \tag{7}$$

This TD algorithm to learn the value and square value functions is a straightforward extension of the algorithm proposed in [23] to the discounted setting. Note that the TD-error $\epsilon$ for the square value function $U$ comes directly from the Bellman equation for $U$ (see Appendix B.1 of [17]).

**(4) Actor Update:** Estimate the gradients $\nabla V^\theta(x^0)$ and $\nabla U^\theta(x^0)$ using SPSA (4) or SF (5) and update the policy parameter $\theta$ and the Lagrange multiplier $\lambda$ as follows: For $i = 1, \ldots, \kappa_1$,

$$\theta_{t+1}^{(i)} = \Gamma_i \left[ \theta_t^{(i)} + \frac{\zeta_2(t)}{\beta \Delta_t^{(i)}} \left( (1 + 2\lambda_t v_t^\top \phi_v(x^0))(v_t^+ - v_t)^\top \phi_v(x^0) - \lambda_t (u_t^+ - u_t)^\top \phi_u(x^0) \right) \right], \textbf{SPSA} \quad (8)$$

$$\theta_{t+1}^{(i)} = \Gamma_i \left[ \theta_t^{(i)} + \frac{\zeta_2(t)\Delta_t^{(i)}}{\beta} \left( (1 + 2\lambda_t v_t^\top \phi_v(x^0))(v_t^+ - v_t)^\top \phi_v(x^0) - \lambda_t (u_t^+ - u_t)^\top \phi_u(x^0) \right) \right], \textbf{SF} \quad (9)$$

$$\lambda_{t+1} = \Gamma_\lambda \left[ \lambda_t + \zeta_1(t) \left( u_t^\top \phi_u(x^0) - \left( v_t^\top \phi_v(x^0) \right)^2 - \alpha \right) \right]. \quad (10)$$

Note that **1)** the $\lambda$-update is the same for both SPSA and SF methods, **2)** $\Delta_t^{(i)}$'s are independent Rademacher and Gaussian $\mathcal{N}(0,1)$ random variables in SPSA and SF updates, respectively, **3)** $\Gamma$ is an operator that projects a vector $\theta \in \mathbb{R}^{\kappa_1}$ to the closest point in a compact and convex set $C \subset \mathbb{R}^{\kappa_1}$, and $\Gamma_\lambda$ is a projection operator to $[0, \lambda_{\max}]$. These projection operators are necessary to ensure convergence of the algorithms, and **4)** the step-size schedules $\{\zeta_3(t)\}$, $\{\zeta_2(t)\}$, and $\{\zeta_1(t)\}$ are chosen such that the critic updates are on the fastest time-scale, the policy parameter update is on the intermediate time-scale, and the Lagrange multiplier update is on the slowest time-scale (see Appendix A of [17] for the conditions on the step-size schedules). A proof of convergence of the SPSA and SF algorithms to a (local) saddle point of the risk-sensitive objective function $\widehat{L}(\theta, \lambda) \triangleq -\widehat{V}^\theta(x^0) + \lambda(\widehat{\Lambda}^\theta(x^0) - \alpha)$ is given in Appendix B.3 of [17].

## 5  Average Reward Setting

The average reward per step under policy $\mu$ is defined as (see Sec. 2 for the definitions of $d^\mu$ and $\pi^\mu$)

$$\rho(\mu) = \lim_{T \to \infty} \frac{1}{T} \mathbb{E} \left[ \sum_{t=0}^{T-1} R_t \mid \mu \right] = \sum_{x,a} d^\mu(x) \mu(a|x) r(x, a).$$

The goal in the standard (risk-neutral) average reward formulation is to find an *average optimal* policy, i.e., $\mu^* = \arg\max_\mu \rho(\mu)$. Here a policy $\mu$ is assessed according to the expected differential reward associated with states or state-action pairs. For all states $x \in \mathcal{X}$ and actions $a \in \mathcal{A}$, the *differential* action-value and value functions of policy $\mu$ are defined as

$$Q^\mu(x, a) = \sum_{t=0}^\infty \mathbb{E} \left[ R_t - \rho(\mu) \mid x_0 = x, a_0 = a, \mu \right], \qquad V^\mu(x) = \sum_a \mu(a|x) Q^\mu(x, a).$$

In the context of risk-sensitive MDPs, different criteria have been proposed to define a measure of *variability*, among which we consider the *long-run variance* of $\mu$ [8] defined as

$$\Lambda(\mu) = \sum_{x,a} \pi^\mu(x, a) \left[ r(x, a) - \rho(\mu) \right]^2 = \lim_{T \to \infty} \frac{1}{T} \mathbb{E} \left[ \sum_{t=0}^{T-1} \left( R_t - \rho(\mu) \right)^2 \mid \mu \right]. \quad (11)$$

This notion of variability is based on the observation that it is the frequency of occurrence of state-action pairs that determine the variability in the average reward. It is easy to show that

$$\Lambda(\mu) = \eta(\mu) - \rho(\mu)^2, \qquad \text{where} \quad \eta(\mu) = \sum_{x,a} \pi^\mu(x, a) r(x, a)^2.$$

We consider the following risk-sensitive measure for average reward MDPs in this paper:

$$\max_\theta \rho(\theta) \qquad \text{subject to} \qquad \Lambda(\theta) \leq \alpha, \quad (12)$$

for a given $\alpha > 0$. As in the discounted setting, we employ the Lagrangian relaxation procedure to convert (12) to the unconstrained problem

$$\max_\lambda \min_\theta \left( L(\theta, \lambda) \triangleq -\rho(\theta) + \lambda \left( \Lambda(\theta) - \alpha \right) \right).$$

Similar to the discounted case, we descend in $\theta$ using $\nabla_\theta L(\theta, \lambda) = -\nabla_\theta \rho(\theta) + \lambda \nabla_\theta \Lambda(\theta)$ and ascend in $\lambda$ using $\nabla_\lambda L(\theta, \lambda) = \Lambda(\theta) - \alpha$, to find the saddle point of $L(\theta, \lambda)$. Since $\nabla \Lambda(\theta) = \nabla \eta(\theta) -$

$2\rho(\theta)\nabla\rho(\theta)$, in order to compute $\nabla\Lambda(\theta)$ it would be enough to calculate $\nabla\eta(\theta)$. Let $U^\mu$ and $W^\mu$ denote the differential value and action-value functions associated with the square reward under policy $\mu$, respectively. These two quantities satisfy the following Poisson equations:

$$\eta(\mu) + U^\mu(x) = \sum_a \mu(a|x)\big[r(x,a)^2 + \sum_{x'} P(x'|x,a)U^\mu(x')\big],$$

$$\eta(\mu) + W^\mu(x,a) = r(x,a)^2 + \sum_{x'} P(x'|x,a)U^\mu(x'). \tag{13}$$

We calculate the gradients of $\rho(\theta)$ and $\eta(\theta)$ as (see Lemma 5 of Appendix C.1 in [17]):

$$\nabla\rho(\theta) = \sum_{x,a} \pi(x,a;\theta)\nabla\log\mu(a|x;\theta)Q(x,a;\theta), \tag{14}$$

$$\nabla\eta(\theta) = \sum_{x,a} \pi(x,a;\theta)\nabla\log\mu(a|x;\theta)W(x,a;\theta). \tag{15}$$

Note that (15) for calculating $\nabla\eta(\theta)$ has close resemblance to (14) for $\nabla\rho(\theta)$, and thus, similar to what we have for (14), any function $b : \mathcal{X} \to \mathbb{R}$ can be added or subtracted to $W(x,a;\theta)$ on the RHS of (15) without changing the result of the integral (see e.g., [2]). So, we can replace $W(x,a;\theta)$ with the square reward advantage function $B(x,a;\theta) = W(x,a;\theta) - U(x;\theta)$ on the RHS of (15) in the same manner as we can replace $Q(x,a;\theta)$ with the advantage function $A(x,a;\theta) = Q(x,a;\theta) - V(x;\theta)$ on the RHS of (14) without changing the result of the integral. We define the temporal difference (TD) errors $\delta_t$ and $\epsilon_t$ for the differential value and square value functions as

$$\delta_t = R(x_t,a_t) - \widehat{\rho}_{t+1} + \widehat{V}(x_{t+1}) - \widehat{V}(x_t), \qquad \epsilon_t = R(x_t,a_t)^2 - \widehat{\eta}_{t+1} + \widehat{U}(x_{t+1}) - \widehat{U}(x_t).$$

If $\widehat{V}, \widehat{U}, \widehat{\rho}$, and $\widehat{\eta}$ are unbiased estimators of $V^\mu, U^\mu, \rho(\mu)$, and $\eta(\mu)$, respectively, then we can show that $\delta_t$ and $\epsilon_t$ are unbiased estimates of the advantage functions $A^\mu$ and $B^\mu$, i.e., $\mathbb{E}[\delta_t | x_t, a_t, \mu] = A^\mu(x_t, a_t)$, and $\mathbb{E}[\epsilon_t | x_t, a_t, \mu] = B^\mu(x_t, a_t)$ (see Lemma 6 in Appendix C.2 of [17]). From this, we notice that $\delta_t\psi_t$ and $\epsilon_t\psi_t$ are unbiased estimates of $\nabla\rho(\mu)$ and $\nabla\eta(\mu)$, respectively, where $\psi_t = \psi(x_t, a_t) = \nabla\log\mu(a_t|x_t)$ is the *compatible* feature (see e.g., [21, 13]).

## 6 Average Reward Algorithm

We now present our risk-sensitive actor-critic algorithm for average reward MDPs. Algorithm 1 presents the complete structure of the algorithm along with update rules for the average rewards $\widehat{\rho}_t, \widehat{\eta}_t$; TD errors $\delta_t, \epsilon_t$; critic $v_t, u_t$; and actor $\theta_t, \lambda_t$ parameters. The projection operators $\Gamma$ and $\Gamma_\lambda$ are as defined in Section 4, and similar to the discounted setting, are necessary for the convergence proof of the algorithm. The step-size schedules satisfy the standard conditions for stochastic approximation algorithms, and ensure that the average and critic updates are on the (same) fastest time-scale $\{\zeta_4(t)\}$ and $\{\zeta_3(t)\}$, the policy parameter update is on the intermediate time-scale $\{\zeta_2(t)\}$, and the Lagrange multiplier is on the slowest time-scale $\{\zeta_1(t)\}$. This results in a three time-scale stochastic approximation algorithm. As in the discounted setting, the critic uses linear approximation for the differential value and square value functions, i.e., $\widehat{V}(x) = v^\top\phi_v(x)$ and $\widehat{U}(x) = u^\top\phi_u(x)$, where $\phi_v(\cdot)$ and $\phi_u(\cdot)$ are feature vectors of size $\kappa_2$ and $\kappa_3$, respectively. Although our estimates of $\rho(\theta)$ and $\eta(\theta)$ are unbiased, since we use biased estimates for $V^\theta$ and $U^\theta$ (linear approximations in the critic), our gradient estimates $\nabla\rho(\theta)$ and $\nabla\eta(\theta)$, and as a result $\nabla L(\theta,\lambda)$, are biased. Lemma 7 in Appendix C.2 of [17] shows the bias in our estimate of $\nabla L(\theta,\lambda)$. We prove that our actor-critic algorithm converges to a (local) saddle point of the risk-sensitive objective function $L(\theta,\lambda)$ (see Appendix C.3 of [17]).

## 7 Experimental Results

We evaluate our algorithms in the context of a traffic signal control application. The objective in our formulation is to minimize the total number of vehicles in the system, which indirectly minimizes the delay experienced by the system. The motivation behind using a risk-sensitive control strategy is to reduce the variations in the delay experienced by road users.

We consider both infinite horizon discounted as well average settings for the traffic signal control MDP, formulated as in [15]. We briefly recall their formulation here: The state at each time $t$, $x_t$, is the vector of queue lengths and elapsed times and is given by $x_t =$

**Algorithm 1** Template of the Average Reward Risk-Sensitive Actor-Critic Algorithm

---

**Input:** parameterized policy $\mu(\cdot|\cdot;\theta)$ and value function feature vectors $\phi_v(\cdot)$ and $\phi_u(\cdot)$
**Initialization:** policy parameters $\theta = \theta_0$; value function weight vectors $v = v_0$ and $u = u_0$; initial state $x_0 \sim P_0(x)$
**for** $t = 0, 1, 2, \ldots$ **do**
    Draw action $a_t \sim \mu(\cdot|x_t;\theta_t)$
    Observe next state $x_{t+1} \sim P(\cdot|x_t, a_t)$
    Observe reward $R(x_t, a_t)$

**Average Updates:** $\quad \widehat{\rho}_{t+1} = \big(1 - \zeta_4(t)\big)\widehat{\rho}_t + \zeta_4(t)R(x_t, a_t), \quad \widehat{\eta}_{t+1} = \big(1 - \zeta_4(t)\big)\widehat{\eta}_t + \zeta_4(t)R(x_t, a_t)^2$

**TD Errors:** $\quad \delta_t = R(x_t, a_t) - \widehat{\rho}_{t+1} + v_t^\top \phi_v(x_{t+1}) - v_t^\top \phi_v(x_t)$

$\qquad\qquad\quad \epsilon_t = R(x_t, a_t)^2 - \widehat{\eta}_{t+1} + u_t^\top \phi_u(x_{t+1}) - u_t^\top \phi_u(x_t)$

**Critic Updates:** $\quad v_{t+1} = v_t + \zeta_3(t)\delta_t\phi_v(x_t), \qquad u_{t+1} = u_t + \zeta_3(t)\epsilon_t\phi_u(x_t)$         (16)

**Actor Updates:** $\quad \theta_{t+1} = \Gamma\Big(\theta_t - \zeta_2(t)\big(-\delta_t\psi_t + \lambda_t(\epsilon_t\psi_t - 2\widehat{\rho}_{t+1}\delta_t\psi_t)\big)\Big)$     (17)

$\qquad\qquad\quad \lambda_{t+1} = \Gamma_\lambda\Big(\lambda_t + \zeta_1(t)(\widehat{\eta}_{t+1} - \widehat{\rho}_{t+1}^2 - \alpha)\Big)$     (18)

**end for**
**return** policy and value function parameters $\theta, \lambda, v, u$

---

$\big(q_1(t), \ldots, q_N(t), t_1(t), \ldots, t_N(t)\big)$. Here $q_i$ and $t_i$ denote the queue length and elapsed time since the signal turned to red on lane $i$. The actions $a_t$ belong to the set of feasible sign configurations. The single-stage cost function $h(x_t)$ is defined as follows:

$$h(x_t) = r_1 \Big[ \sum_{i \in I_p} r_2 \cdot q_i(t) + \sum_{i \notin I_p} s_2 \cdot q_i(t) \Big] + s_1 \Big[ \sum_{i \in I_p} r_2 \cdot t_i(t) + \sum_{i \notin I_p} s_2 \cdot t_i(t) \Big], \qquad (19)$$

where $r_i, s_i \geq 0$ such that $r_i + s_i = 1$ for $i = 1, 2$ and $r_2 > s_2$. The set $I_p$ is the set of prioritized lanes in the road network considered. While the weights $r_1, s_1$ are used to differentiate between the queue length and elapsed time factors, the weights $r_2, s_2$ help in prioritization of traffic.

Given the above traffic control setting, we aim to minimize both the long run discounted as well average sum of the cost function $h(x_t)$. The underlying policy for all the algorithms is a parameterized Boltzmann policy (see Appendix F of [17]). We implement the following algorithms in the discounted setting:
**(i)** Risk-neutral SPSA and SF algorithms with the actor update as follows:

$$\theta_{t+1}^{(i)} = \Gamma_i \left( \theta_t^{(i)} + \frac{\zeta_2(t)}{\beta\Delta_t^{(i)}}(v_t^+ - v_t)^\top\phi_v(x^0) \right) \quad \textbf{SPSA},$$

$$\theta_{t+1}^{(i)} = \Gamma_i \left( \theta_t^{(i)} + \frac{\zeta_2(t)\Delta_t^{(i)}}{\beta}(v_t^+ - v_t)^\top\phi_v(x^0) \right) \quad \textbf{SF},$$

where the critic parameters $v_t^+, v_t$ are updated according to (6). Note that these are two-timescale algorithms with a TD critic on the faster timescale and the actor on the slower timescale.
**(ii)** Risk-sensitive SPSA and SF algorithms (RS-SPSA and RS-SF) of Section 4 that attempt to solve (2) and update the policy parameter according to (8) and (9), respectively. In the average setting, we implement **(i)** the risk-neutral AC algorithm from [14] that incorporates an actor-critic scheme, and **(ii)** the risk-sensitive algorithm of Section 6 (RS-AC) that attempts to solve (12) and updates the policy parameter according to (17).

All our algorithms incorporate function approximation owing to the curse of dimensionality associated with larger road networks. For instance, assuming only 20 vehicles per lane of a 2x2-grid network, the cardinality of the state space is approximately of the order $10^{32}$ and the situation is aggravated as the size of the road network increases. The choice of features used in each of our algorithms is as described in Section V-B of [16]. We perform the experiments on a 2x2-grid network. The list of parameters and step-sizes chosen for our algorithms is given in Appendix F of [17].

Figures 2(a) and 2(b) show the distribution of the discounted cumulative reward $D^\theta(x^0)$ for the SPSA and SF algorithms, respectively. Figure 3(a) shows the distribution of the average reward $\rho$ for the algorithms in the average setting. From these plots, we notice that the risk-sensitive algorithms

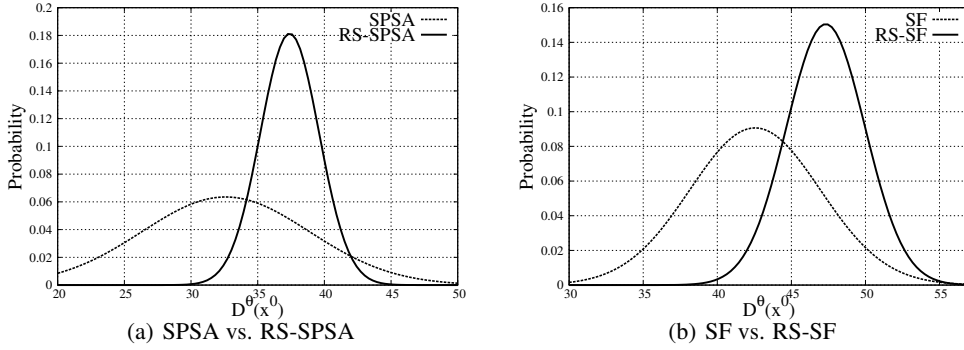

(a) SPSA vs. RS-SPSA        (b) SF vs. RS-SF

Figure 2: Performance comparison in the discounted setting using the distribution of $D^\theta(x^0)$.

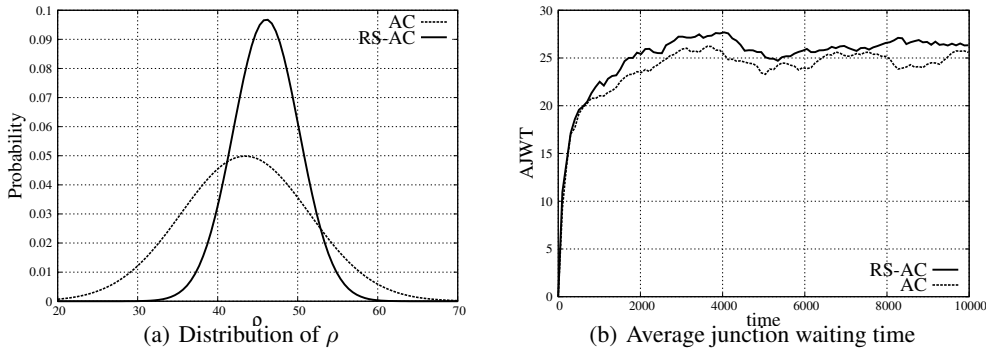

(a) Distribution of $\rho$        (b) Average junction waiting time

Figure 3: Comparison of AC vs. RS-AC in the average setting using two different metrics.

that we propose result in a long-term (discounted or average) reward that is higher than their risk-neutral variants. However, from the empirical variance of the reward (both discounted as well as average) perspective, the risk-sensitive algorithms outperform their risk-neutral variants.

We use average junction waiting time (AJWT) to compare the algorithms from a traffic signal control application standpoint. Figure 3(b) presents the AJWT plots for the algorithms in the average setting (see Appendix F of [17] for similar results for the SPSA and SF algorithms in the discounted setting). We observe that the performance of our risk-sensitive algorithms is not significantly worse than their risk-neutral counterparts. This coupled with the observation that our algorithms exhibit low variance, makes them a suitable choice in risk-constrained systems.

## 8   Conclusions and Future Work

We proposed novel actor critic algorithms for control in risk-sensitive discounted and average reward MDPs. All our algorithms involve a TD critic on the fast timescale, a policy gradient (actor) on the intermediate timescale, and dual ascent for Lagrange multipliers on the slowest timescale. In the discounted setting, we pointed out the difficultly in estimating the gradient of the variance of the return and incorporated simultaneous perturbation based SPSA and SF approaches for gradient estimation in our algorithms. The average setting, on the other hand, allowed for an actor to employ compatible features to estimate the gradient of the variance. We provided proofs of convergence (in the appendix of [17]) to locally (risk-sensitive) optimal policies for all the proposed algorithms. Further, using a traffic signal control application, we observed that our algorithms resulted in lower variance empirically as compared to their risk-neutral counterparts.

In this paper, we established asymptotic limits for our discounted and average reward risk-sensitive actor-critic algorithms. To the best of our knowledge, there are no convergence rate results available for multi-timescale stochastic approximation schemes and hence for actor-critic algorithms. This is true even for the actor-critic algorithms that do not incorporate any risk criterion. It would be an interesting research direction to obtain finite-time bounds on the quality of the solution obtained by these algorithms.

## Footnotes

[1]We note here that our algorithms can be easily extended to other variance-related risk criteria such as the Sharpe ratio, which is popular in financial decision-making [18] (see Appendix D of [17]).

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
