[Reviews · NeurIPS 2013]

Submitted by Assigned_Reviewer_5

The paper proposes a novel actor-critic algorithm for risk-sensitive MDPs - both for the discounted and the average reward case.


Clarity
-------
The paper is very well written, logical and properly structured.
The appendices are mathematically intense and demands significant effort to follow.
The quality of English is excellent.


Quality
-------
The paper is very rigorous, the descriptions are detailed, and the methodology used for the derivations is sound. The title is a good match for the content.

There is one issue that the authors do not discuss: how to select a good value for \alpha in (2) and (12)? In Appendix F the authors state that they used value \alpha = 20, but there is not discussion about how to choose this threshold.

In Figure 2, it is not clear to me why the average junction waiting time is increasing instead of decreasing as a result of the optimization?


Originality
-----------
Section 1 does a perfect job at positioning the paper's contribution with respect to existing prior work. The proposed method is fairly original and theoretically sound.

The evaluation part is a bit weak, because only one traffic simulation was performed on a small-scale 2x2 grid which is not that convincing.
Overall, the novelty is at a satisfactory level.


Significance
------------
In my opinion, the work is very good and thorough, and presents a worthwhile advancement over the state of the art.


Summary: The paper proposes a novel actor-critic algorithm for risk-sensitive MDPs together with convergence proofs and detailed derivations. Thus, it is a worthwhile advancement over the state of the art.

Submitted by Assigned_Reviewer_6

The authors present actor-critic algorithms for learning risk-sensitive policies. They present infinite-horizon methods for both the discounted reward and average reward cases, and present empirical results demonstrating their use on a simulated traffic control problem.

-Quality: The paper appears to be sound and thorough. It addresses an important problem.
-Clarity: The paper is clear; I have suggestions below.
-Originality: To the best of my knowledge, the approach and results are novel.
-Significance: The paper improves our understanding and ability to deal with risk in RL, which is important for many problems.

----Specific suggestions and questions:
-"variance related" ... "variance-related"
-Don't use parenthetical references as nouns. ("similar to [21]...")
-"...lacks the monotonicity property..." I think a cite here or an explanation of what "monotonicity" means in this setting is needed.
-"...in order to compute \nabla\Lambda^\theta(x^0), it would be enough to calculate \nabla{}U^\theta(x^0)." I found this confusing; I think you mean because we need to compute \nabla{}V anyway? I think a more clear statement about how to go beyond "traditional" actor-critic we need only this additional term would help, perhaps.
-Lemma 1 does not fit into the column width.
-\kappa_1 is used before definition; mention what space \theta lives in first in order for things to make sense.
-In (5), \theta^i should be \theta^{(i)}
-In the SPSA case, since \Delta^{(i)} are Rademacher, they could equally be written in the numerator. I understand this is not quite "in the spirit" of the derivative approximation, where the step-size would appear in the denominator, but I think the win of being able to re-write (4) and (5) in exactly the same way makes it worth it. Then you don't have to repeat yourselves later on, which would free up some space.
-Do you care what norm the projection operator is in? It would be worth mentioning either way.

-I think there is a missed opportunity to discuss why the discounted reward case or the average reward case is more appropriate for a particular problem. In the average-reward case, the variance of R(s,a)---that is, the variance of the reward random variable at a fixed state-action pair---does not impact the objective, whereas in the discounted case it does. This seems like it might matter. Can you comment on why one might prefer one over the other?

-Related, rather than "...it is the frequency of occurrence of state-action pairs that determine the variability in the average reward." I might say it is *only* the variability in frequency of occurrence.

-I think \cdot (or nothing at all) would be better than * in (19)
-"...is not significantly low..." I would say "poor," since lower is better in this case; furthermore avoid the word significant unless you will explain that you mean statistical significance, practical significance, or both.
-Give plots in Figure 2 the same x-range.
-RS-SF looks like it dominates RS in Figure 2b. Does it? Why might this be? ADDENDUM: Is this reward or cost? If larger quantities are not better, *do not call it reward*. Either negate it or change terminology.

-Assuming it is "cost" rather than reward, it looks like SF returns might be stochastically smaller than RS-SF, even though it has larger variance. In this case, why should the RS-SF policy be desirable? Is it? In general, the argument that the reduction in variance is "worth it" is not very convincing. Perhaps explain what the units mean and that the difference is not really relevant.

-"difficultly" -- Spellcheck your submission.
-Hyphenate before -based. "perturbation-based, and so on."
Summary: The authors present actor-critic algorithms for learning risk-sensitive policies. They clearly present the new algorithms, describe their properties, and demonstrate their use on an interesting problem.

Submitted by Assigned_Reviewer_7

The paper addresses the problem of finding a policy with a high expected return and a bounded variance. The paper considers both the discounted and the average reward cases. The authors propose formulate this problem as a constrained optimization problem, where the gradient of the Lagrangian dual function is estimated form samples. This gradient is composed of the gradient of the expected return and the gradient of the expected squared return. Both gradients need to be estimated in every state. The authors use a linear function approximation to generalize the gradient estimates to states that were not encountered in the samples. The authors use stochastic perturbation to evaluate the gradients in particular states by sampling two trajectories, one with policy parameters theta and another with policy parameters theta+beta, where beta is a perturbation random variable. The policy parameters are updated in an actor-critic scheme. The authors prove that the proposed optimization method converges to a local optimum. Numerical experiments on a traffic lights control problem show that the proposed technique finds a policy with a slightly higher risque than the optimal solution, but with a significantly lower variance.

Quality:
This work is of a good quality, the derivations seem correct and the empirical results confirm the claims made about the proposed technique.

Clarity:
The paper is clear. One point that was not clear to me is how the Bellman Equations in Lemma 1 were used for calculating the TD updates. The authors explained this point in the appendix, but it should be moved to the paper.

Originality:
Although the work mostly builds on Tamar et al. 2012, the proof of convergence using three updates scales in a stochastic optimization seems quite new here. The use of linear function approximation to generalize the estimates of the gradient (not only the value) seems also new. I also find the use of stochastic perturbation appealing in this context.

Significance:
The numerical experiments show that the proposed method can be useful in practical scenarios, where one needs to control the variance of the solution. Therefore, I consider this work significant.

Questions:
- How do you account for the variance resulting from the stochastic perturbation? Does your method have a high-variance learning phase?
- The explanations provided after Equations (13) and (14) are not consistent with the equations.
- It seems like most of the variance of the non-risk-averse policy in the experiments is around low risk regions, which is a good thing. For instance, in Figure 2.b the non-risk averse policy has a zero probability of reaching high cost values, which is not the case of the risk-averse policy.
Summary: The paper solves an important problem related to risk averse sequential decision making, the proposed method seems novel and convincing.
Author Feedback

Author rebuttal: We would like to thank the reviewers for their useful comments.

Remark: Managing risk in decision problems is extremely important in many different fields. Now that the theory of risk-sensitive MDPs is relatively well-understood and that we know many of such problems are computationally intractable, it is important to develop algorithms, with convergence and (possibly) performance guarantees, for approximately solving these problems. Unfortunately there has not been many work in this area (approximately solving risk-sensitive MDPs), and thus, we believe any attempt in this direction could potentially have a major impact.


Reviewer 1:
----------

- The choice of \alpha depends on the amount of variability (in the performance) that can be tolerated and hence is application dependent. In the traffic signal control application, we observed the mean value to be approximately 40 and thus we chose \alpha as 20.

- With respect to the comment regarding the average junction waiting time (AJWT), we believe you are referring to Fig 3(b). Since the risk sensitive RS-AC algorithm is attempting to solve a constrained problem, we expect a decay in the performance and Fig 3(b) shows that the decay is not significant in comparison to plain actor-critic algorithm. Further, this observation should be seen in conjunction with Fig 3(a) which shows that RS-AC results in lower variance. Also, the increase in AJWT initially is owing to the fact that the simulation starts with zero vehicles and vehicles get added with time according to a Poisson distribution. The AJWT plots show that our algorithms stabilize the AJWT and the transient (initial) period is short.

- We shall incorporate all the useful suggestions in the final version of the paper.


Reviewer 2:
----------
- We agree that it is useful to merge Eq. (4) and (5) considering that \Delta^{(i)} are Rademacher. The reason we kept it separate was due to the fact that more general (non-Rademacher) distributions could be used for \Delta^{(i)} in the SPSA estimate.

- In response to the question regarding the choice of setting - discounted or average, we believe that this choice is motivated by considerations as in the risk-neutral case. In other words, discounted setting is useful for studying the transient behavior of the system, whereas the average setting is to understand steady-state system behavior. Further, in this paper, we defined risk in a manner specific to the setting considered.

- We would like to clarify that the experimental setting involves "cost" and not reward and we shall clarify the terminology in the final version of the paper. Fig 3(b) of the main paper and Figs. 2 and 3 in the appendix plot the average junction waiting time while Figs. 2(a)-(b) and 3(a) plot the distribution of the return (\rho in the average and D^\theta(x^0) in the discounted settings). Empirically we observe that the risk sensitive algorithms result in lower variance but higher return (cost). Further, from a average junction waiting time perspective, the risk sensitive algorithms' performance is close to their risk-neutral counterparts, thus making them amenable for use in risk constrained systems.

- We shall incorporate all the useful suggestions and correct all the minor errors in the final version of the paper.


Reviewer 3:
----------

- We agree that TD updates using Lemma 1 can be made clearer, perhaps by moving some content from the appendix and this shall be done in the final version of the paper.

- In response to the question regarding variance of the perturbation based gradient estimates, we would like to clarify that it is difficult to analyze it in theory. However, empirically, from results averaged our 100 independent simulation runs we observed that our algorithms did not exhibit a high-variance learning phase.

- We shall incorporate all the useful suggestions in the final version of the paper.